# Bridging Extremes: The Invertible Bimodal Gumbel Distribution

**DOI:** 10.3390/e25121598

**Published:** 2023-11-29

**Authors:** Cira G. Otiniano, Eduarda B. Silva, Raul Y. Matsushita, Alan Silva

**Affiliations:** 1Department of Statistics, University of Brasília, Brasília 70910-900, Brazil; eduarda.silva@brb.com.br (E.B.S.); raulmta@unb.br (R.Y.M.); 2Institute of Mathematics and Statistics, São Paulo University, São Paulo 05508-220, Brazil; alansilva@ime.usp.br

**Keywords:** Gumbel distribution, bimodality, extreme value theory, value at risk

## Abstract

This paper introduces a novel three-parameter invertible bimodal Gumbel distribution, addressing the need for a versatile statistical tool capable of simultaneously modeling maximum and minimum extremes in various fields such as hydrology, meteorology, finance, and insurance. Unlike previous bimodal Gumbel distributions available in the literature, our proposed model features a simple closed-form cumulative distribution function, enhancing its computational attractiveness and applicability. This paper elucidates the behavior and advantages of the invertible bimodal Gumbel distribution through detailed mathematical formulations, graphical illustrations, and exploration of distributional characteristics. We illustrate using financial data to estimate Value at Risk (VaR) from our suggested model, considering maximum and minimum blocks simultaneously.

## 1. Introduction

Bimodal heavy-tailed distributions are powerful analytical tools for capturing the complex nature of phenomena subject to extreme events in hydrology, meteorology, insurance, reliability, and finance, among other disciplines. Two main features characterize it. Firstly, it exhibits bimodality, meaning it has two distinct peaks or modes, indicating the presence of two prominent regimes within the overall data set. Secondly, it has heavy tails, which means a higher likelihood of occurrences of extreme values than with light tails distributions.

Bimodal heavy-tailed distributions are related to René Thom’s catastrophe theory, focusing on systems characterized by sudden, dramatic changes and a propensity for extreme events, e.g., [1,2]. Catastrophe theory deals with situations where small parameter changes can lead to abrupt shifts in the system’s state. This concept aligns with bimodal distributions, where a system may switch between two states or regimes. The heavy-tailed aspect of these distributions reflects the likelihood of rare, extreme events, mirroring the focus of catastrophe theory on significant, sudden changes. Both concepts encapsulate the unpredictability and uncertainty inherent in the systems they describe. Catastrophe theory provides a mathematical framework for understanding these dynamics, which can manifest statistically as bimodal, heavy-tailed distributions. This connection is especially relevant in economics and finance, where catastrophic shifts and heavy-tailed distributions are frequently observed. Essentially, the interplay between these concepts helps us to understand and model systems where small inputs or changes can lead to significant, unpredictable, and often extreme outputs or shifts.

Unlike standard bimodal distributions, these heavy-tailed versions give significant weight to extreme events, allowing for a more accurate representation of systems where outliers or “black swans” play a critical role. For example, one can perform inference over tails of financial returns by fitting an appropriate limiting distribution over data that exceeds a fixed threshold, where the dual peaks in such a distribution can indicate states or types of behavior within the system [3]. In addition, typically, a bimodal distribution exhibits higher entropy compared to an unimodal distribution because two distinct modes add complexity and unpredictability to the system. In this way, the concept of entropy dovetails nicely with its inherent complexities, providing a quantitative lens through which to assess and strategize based on this kind of data.

Meanwhile, under certain conditions, statistics of extreme events are described by theoretical distributions. For example, the Gumbel, also known as the extreme value or Fisher–Tippet type I distribution, is a limiting distribution for the maximum (or the minimum) of a sufficiently large simple random sample. This result arises from the Fisher–Tippett–Gnedenko theorem, which states that the normalized maximum of a sequence of such random samples converges to one of three types of extreme value distribution: Gumbel, Fréchet, or Weibull. The Gumbel case is suitable for some typical families of populational distributions, such as logistic, Gaussian, and gamma.

Nevertheless, practical situations demand more, and therefore we find several generalizations of Gumbel to make it more flexible, for example, the two-component extreme value distribution or mixture of two Gumbel distributions [4], the exponentiated Gumbel [5], the transmuted extreme value [6], the generalized Gumbel [7], the generalized three-parameter Gumbel [8], the beta-Gumbel [9], the Kumaraswamy-Gumbel [10], and the exponentiated generalized Gumbel [11]. However, some lead to non-identifiable models because of observationally equivalent parameterizations [12]. There are other closely related models such as the exponentiated Gumbel Type-2 [13], the Kumaraswamy generalized exponentiated Gumbel type-2 [14], the bimodal generalized extreme value (GEV) [15], and a bimodal Gumbel distribution applied to environmental data [16]. However, the disadvantage of the latter model is that its cumulated distribution function does not have a simple closed form.

In this work, we put forward an invertible bimodal Gumbel distribution whose cumulated distribution function has a simple closed form, making it more attractive for computational procedures and more flexible in applications (Section 2). Our suggested distribution allows us to model both maximum and minimum simultaneously, while the classical Gumbel distribution describes only one of the extremes (maximum or minimum).

After discussing the maximum likelihood estimation of the parameters from simulated data (Section 3), we illustrate our approach using two financial data sets to estimate the value at risk (VaR) in Section 4. As we are interested in studying the probability distribution’s tails, we perform the block maxima technique among the available tools to find the appropriate cutoff. Instead of the usual power law distribution [3,17,18], we suggest a bimodal Gumbel distribution as a candidate model to describe the tail behavior of financial returns.

## 2. Main Results

Also known as type I extreme value distribution, the Gumbel distribution is one of the limit distributions of normalized maximum (or minimum) statistics [19], belonging to the class of the GEV distribution [20]. We denote the Gumbel random variable *Y* with a location parameter μ∈R and a scale parameter σ>0 as Y∼G(·;μ,σ). The forms of its probability density function (PDF) and cumulative distribution function (CDF) are, respectively,
(1)$$\begin{array}{c} g\left(y;\mu ,\sigma \right)=\frac{1}{\sigma }\exp\left\{-\left(\frac{y-\mu }{\sigma }\right)-\exp\left[-\left(\frac{y-\mu }{\sigma }\right)\right]\right\},\;\forall y\in R \end{array}$$
and
(2)$$\begin{array}{cc} G(y;\sigma ,\mu ) & =\exp\left[-\exp\left(-\frac{y-\mu }{\sigma }\right)\right],\;\forall y\in R. \end{array}$$

Let us introduce our suggested generalization of the Gumbel distribution left open by [15] in the following way. Considering the transformation,
(3)$$\begin{array}{c} T_{\mu ,\delta }\left(x\right)=x{\left|x\right|}^{\delta }+\mu \;,\;x\in R,\;\;\delta >-1,\;\;\mu \in R, \end{array}$$
after plugging it into (Equation 1) and (Equation 2), we obtain the invertible bimodal Gumbel distribution *X* with CDF and PDF given by, respectively,
(4)$$\begin{array}{ccc} F_{\mathrm{IBG}}\left(x;\mu ,\sigma ,\delta \right) & = & G\left(T_{\mu ,\delta }\left(x\right);\sigma \right)\\ & = & \exp\left\{-\exp\left[-\frac{(x|x{|}^{\delta })+\mu }{\sigma }\right]\right\},\;x\in R \end{array}$$
and
(5)$$\begin{array}{ccc} f_{\mathrm{IBG}}\left(x;\mu ,\sigma ,\delta \right) & = & \frac{1}{\sigma }{\left(\delta +1\right)|x|}^{\delta }\exp\left\{-\left(\frac{(x|x{|}^{\delta })+\mu }{\sigma }\right)-\exp\left[-\frac{(x|x{|}^{\delta })+\mu }{\sigma }\right]\right\}, \end{array}$$
where δ>0 and μ∈R are shape parameters and σ>0 is a scale parameter.

We shall denote it as $X∼F_{\mathrm{IBG}}\left(\cdot ;\mu ,\sigma ,\delta \right)$ throughout this paper.

To illustrate the role of its parameters, Figure 1 depicts the effect of the shape parameter δ. When δ=0, the model (Equation 5) reduces to the unimodal Gumbel (Equation 1). The density becomes bimodal for δ>0, and the modes’ separation rises as δ increases. Figure 2 contrasts the PDF shapes with negative and positive values of μ, illustrating its role as a location parameter (left) or shape parameter (right). Finally, Figure 3 shows that σ remains the scale parameter.

### Some Distributional Characteristics

**Modes 1.** Straightforwardly from the concept of the modes of $X{∼}_{GB}\left(\cdot ;\mu ,\sigma ,\delta \right)$, one can find that they are the solution of the differential equation
(6)$$\begin{array}{c} \frac{T_{\mu ,\delta }^{\prime \prime }\left(x\right)}{{\left[T_{\mu ,\delta }^{\prime }\left(x\right)\right]}^{2}}=\frac{1}{\sigma }-\frac{e^{-\frac{T_{\mu ,\delta }\left(x\right)}{\sigma }}}{\sigma }, \end{array}$$
where
(7)$$\begin{array}{c} T_{\mu ,\delta }^{\prime }\left(x\right)=\left(\delta +1\right){\left|x\right|}^{\delta } \end{array}$$
and
(8)$$\begin{array}{c} T_{\mu ,\delta }^{\prime \prime }\left(x\right)=\mathrm{sign}\left(x\right)\;\left(\delta +1\right)\delta {\left|x\right|}^{\delta -1}\;, \end{array}$$
with sign(x)=x/|x| as the sign function.

**Moments 2.** We can write down the *k*th moment of *X* as
(9)$$\begin{array}{ccc} E(X^{k}) & = & \int_{-\infty }^{\infty }x^{k}f_{\mathrm{IBG}}\left(x\right)dx\\ & = & \int_{-\infty }^{\infty }x^{k}g\left(T_{\mu ,\delta }\left(x\right),0,\mu \right)T_{\mu ,\delta }^{\prime }\left(x\right)dx. \end{array}$$
By substitution y=T(x) and taking the inverse function $T^{-1}\left(y\right)={sgn(|y-\mu |)|y-\mu |}^{\frac{1}{\delta +1}}$, we can express (Equation 9) in terms of a unimodal Gumbel *Y*, as defined in (Equation 2) as
(10)$$\begin{array}{c} E\left(X^{k}\right)=E\left({\left|Y-\mu \right|}^{\frac{k}{\delta +1}}.I_{\left[\mu ,+\infty \right)}\right)+{\left(-1\right)}^{k}E\left({\left|Y-\mu \right|}^{\frac{k}{\delta +1}}.I_{\left(-\infty ,\mu \right)}\right), \end{array}$$
where $I_{A}$ is the indicator function of an event *A*.

**Moment-Generating Type Function 3.** The moment-generating function (MGF) encapsulates information about the distributional moments, being a helpful tool to characterize an IBG random variable *X*. For our convenience, however, we consider its power transformation $X^{\delta +1}$ and derive its MGF shown in (Equation 13) as follows. From its definition,
(11)$$\begin{array}{ccc} \varphi_{X}\left(t\right)=E\left(e^{tX^{\delta +1}}\right) & = & \int_{-\infty }^{\infty }\frac{e^{tx^{1+\delta }}}{\sigma }e^{\frac{-T(x)}{\sigma }}e^{-e^{\frac{-T(x)}{\sigma }}}T^{\prime }\left(x\right)dx. \end{array}$$
Considering the substitution $y=e^{\frac{-T(x)}{\sigma }}$ and the expression $T^{-1}\left(\ln y^{-\sigma }\right)=sgn(|\ln y^{-\sigma }-\mu |)|\ln y^{-\sigma }{-\mu |}^{\frac{1}{\delta +1}}$, we can rewrite the integral (Equation 11) as being
(12)$$\begin{array}{ccc} \varphi_{X}\left(t\right) & = & \int_{0}^{e^{\frac{\mu }{\sigma }}}e^{t(-\sigma \ln y-\mu )}e^{-y}dy+\int_{e^{\frac{\mu }{\sigma }}}^{\infty }e^{{\left(-1\right)}^{1+\delta }t(\mu +\sigma \ln y)}e^{-y}dy\\ & = & e^{-t\mu }\int_{0}^{e^{\frac{\mu }{\sigma }}}y^{-t\sigma }e^{-y}dy+e^{{\left(-1\right)}^{1+\delta }\left(t\mu \right)}\int_{e^{\frac{\mu }{\sigma }}}^{\infty }y^{t\sigma {\left(-1\right)}^{1+\delta }}e^{-y}dy, \end{array}$$
that is,
(13)$$\begin{array}{c} \varphi_{X}\left(t\right)=e^{-t\mu }\Gamma \left(1-t\sigma ;e^{\frac{\mu }{\sigma }}\right)+e^{t\mu {\left(-1\right)}^{\delta +1}}\gamma \left(1+{\left(-1\right)}^{\delta +1}t\sigma ;e^{\frac{\mu }{\sigma }}\right), \end{array}$$
where
(14)$$\begin{array}{c} \gamma \left(a;x\right)=\int_{x}^{\infty }t^{a-1}e^{-t}dt\;\;\;\mathrm{and}\;\;\;\Gamma \left(a;x\right)=\int_{0}^{x}t^{a-1}e^{-t}dt, \end{array}$$
are the upper and lower incomplete Gamma functions. Now, we can retrieve the moments of $X^{\delta +1}$ by taking derivatives of the cumulant-generating type function, $C_{X}=\ln\varphi_{X}\left(t\right)$. As usual, from the expansion $\ln z=\left(z-1\right)-{\left(z-1\right)}^{2}/2+{\left(z-1\right)}^{3}/3-\ldots$, we find
(15)$$\begin{array}{c} C_{X}\left(t\right)=\left(E\left(e^{tX^{\delta +1}}\right)-1\right)-\frac{{\left(E\left(e^{tX^{\delta +1}}\right)-1\right)}^{2}}{2}+\frac{{\left(E\left(e^{tX^{\delta +1}}\right)-1\right)}^{3}}{3}-\ldots \end{array}$$
Thus, for example, we get the first two moments of $X^{\delta +1}$ by taking the derivatives
(16)$$\begin{array}{ccc} \frac{d}{dt}C_{X}\left(t\right){|}_{t=0} & = & E(X^{\delta +1})\\ \frac{d^{2}}{dt^{2}}C_{X}\left(t\right){|}_{t=0} & = & E\left(X^{2(\delta +1)}\right)-{\left[E\left(X^{\delta +1}\right)\right]}^{2}.. \end{array}$$
To benchmark our result, if δ=0, from (Equation 13) and (Equation 16), we find E(X)=σγ−μ and $Var\left(X\right)=\sigma^{2}\pi^{2}/6$, because Γ(a;x)+γ(a;x)=Γ(a), where $\Psi \left(x\right)=d\ln\left(\Gamma \left(x\right)\right)/dx=\Gamma^{\prime }\left(x\right)/\Gamma \left(x\right)$, Ψ(1)=γ is the Euler’s constant, and $\Psi^{\prime }\left(1\right)=\frac{\pi^{2}}{6}$. Thus, we get the mean and variance of the basic Gumbel model as expected, which confirms the accuracy of (Equation 13).

**Quantiles 4.** Sampling by the inverse transform is a basic method with which to generate a pseudo-random variate of *X*, based on its quantile function of $F_{IBG}$. While the bimodal Gumbel model introduced previously by [16] does not provide a simple way to perform this method, our suggested model (Equation 5) yields a simple expression for the quantile function. Since *X* is an absolutely continuous random variable, denoting the cumulative probability as the standard uniform random variable $F_{IBG}\left(x_{q};\mu ,\sigma ,\delta \right)=q∼U\left[0,1\right]$, we obtain the random quantile function as
(17)$$\begin{array}{ccc} X_{q}=F_{IBG}^{-1}\left(q\right) & = & \left\{\begin{array}{cc} {\left(-\sigma \ln(-\ln(q))-\mu \right)}^{\frac{1}{1+\delta }}, & q>e^{-e^{-\frac{\mu }{\sigma }}},\\ -{\left(\mu +\sigma \ln(-\ln(q))\right)}^{\frac{1}{1+\delta }}, & q<e^{-e^{-\frac{\mu }{\sigma }}}. \end{array}\right. \end{array}$$

**Entropy 5.** The differential entropy of the bimodal Gumbel distribution $X∼F_{IBG}\left(\cdot ;\mu ,\sigma ,\delta \right)=G\left(Y\left(T_{\mu ,\delta }\left(.\right),\sigma \right)\right)$, where Y∼G(.,σ) denotes the basic Gumbel distribution, is given by
(18)$$\begin{array}{ccc} H(X) & = & -\int_{-\infty }^{\infty }f_{IBG}\left(x;\mu ,\sigma ,\delta \right)\ln\left[f_{IBG}\left(x;\mu ,\sigma ,\delta \right)\right]dx\\ & = & \int_{-\infty }^{\infty }g\left(T_{\mu ,\delta }\left(x\right);\sigma \right)\left\{\exp\left[-\frac{T_{\mu ,\delta }\left(x\right)}{\sigma }\right]+\frac{T_{\mu ,\delta }\left(x\right)}{\sigma }+\ln\left(\frac{T_{\mu ,\delta }^{\prime }\left(x\right)}{\sigma }\right)\right\}\frac{T_{\mu ,\delta }^{\prime }\left(x\right)}{\sigma }dx\\ & = & \int_{-\infty }^{\infty }g\left(T_{\mu ,\delta }\left(x\right);\sigma \right)\exp\left[-\frac{T_{\mu ,\delta }\left(x\right)}{\sigma }\right]\frac{T_{\mu ,\delta }^{\prime }\left(x\right)}{\sigma }dx+\int_{-\infty }^{\infty }g\left(T_{\mu ,\delta }\left(x\right);\sigma \right)\frac{T_{\mu ,\delta }\left(x\right)}{\sigma }\frac{T_{\mu ,\delta }^{\prime }\left(x\right)}{\sigma }dx\\ & + & \int_{-\infty }^{\infty }g\left(T_{\mu ,\delta }\left(x\right);\sigma \right)\ln\left(\frac{T_{\mu ,\delta }^{\prime }\left(x\right)}{\sigma }\right)\frac{T_{\mu ,\delta }^{\prime }\left(x\right)}{\sigma }dx, \end{array}$$
where *g* is the PDF of *Y*, as defined in (Equation 2). By substituting y=T(x) in (Equation 18), we obtain
(19)$$\begin{array}{ccc} H(X) & = & \frac{1}{\sigma }\int_{-\infty }^{\infty }\exp\left[-\frac{y}{\sigma }\right]g\left(y\right)dy+\frac{1}{\sigma^{2}}\int_{-\infty }^{\infty }yg\left(y\right)dy\\ & + & \frac{1}{\sigma }\int_{-\infty }^{\infty }\ln\left(\frac{\delta +1}{\sigma }\right)g\left(y\right)dy+\frac{1}{\sigma }\int_{-\infty }^{\infty }\ln\left({\left|y-\mu \right|}^{\frac{\delta }{\delta +1}}\right)dy\\ & = & 1+\frac{\gamma }{\sigma }+\ln{\left(\frac{\delta +1}{\sigma }\right)}^{\frac{1}{\sigma }}+\ln\left({E|Y-\mu |}^{\frac{\delta }{\delta +1}}\right). \end{array}$$

## 3. Parameter Estimation

This section discusses the maximum likelihood (ML) estimation method to estimate the vector parameters Θ=(μ,σ,δ). Let $x_{1},\cdots ,x_{n}$ be realizations independent copies of a random variable with PDF as defined in (Equation 5). The log-likelihood function is
(20)$$\begin{array}{ccc} l(\Theta ;x_{1},x_{2},\cdots ,x_{n}) & = & \sum_{i=1}^{n}\ln f\left(x_{i};\Theta \right)\\ & = & n\ln\left(\delta +1\right)-n\ln\sigma +\delta \sum_{i=1}^{n}\ln\left|x_{i}\right|-\frac{\sum_{i=1}^{n}x_{i}{\left|x_{i}\right|}^{\delta }+\mu }{\sigma }\\ & - & \sum_{i=1}^{n}e^{-\frac{x_{i}{\left|x_{i}\right|}^{\delta }+\mu }{\sigma }}. \end{array}$$

This log-likelihood function is well-defined across the entire parameter space and is continuous and differentiable for the vector parameters. Additionally, the family of distributions $F_{\mathrm{IBG}}$ is identifiable, meaning different parameters should lead to distinct probability distributions, ensuring a unique maximum for the likelihood function.

Ahmad et al. (2010) [21] showed the identifiability of the finite mixture of Gumbel distributions; in particular, the family of a Gumbel component $F_{G}=\left\{G:G=G\left(.,\mu ,\sigma \right)\;\mathrm{as}\;\left(2\right)\right\}$ is identifiable. Based on this, we have that the IBG family, $F_{\mathrm{IBG}}=\left\{F_{\mathrm{IBG}}:F_{\mathrm{IBG}}\left(.,\mu ,\sigma ,\delta \right)\;\mathrm{as}\;\left(4\right)\right\}$, is identifiable. It must be proven that
$$\begin{array}{c} F_{\mathrm{IBG}}\left(x;\mu_{1},\sigma_{1},\delta_{1}\right)=F_{\mathrm{IBG}}\left(x;\mu_{2},\sigma_{2},\delta_{2}\right)\;\;\mathrm{if}\;\mathrm{and}\;\mathrm{only}\;\mathrm{if}\;\;\mu_{1}=\mu_{2},\sigma_{1}=\sigma_{2},\delta_{1}=\delta_{2}. \end{array}$$

Indeed, from (Equation 4)
(21)$$\begin{array}{c} \exp\left\{-\exp\left[-\frac{{x|x|}^{\delta_{1}}+\mu_{1}}{\sigma_{1}}\right]\right\}=\exp\left\{-\exp\left[\frac{{x|x|}^{\delta_{2}}+\mu_{2}}{\sigma_{2}}\right]\right\}. \end{array}$$
Since $F_{G}$ is identifiable, then $\mu_{1}=\mu_{2}$ and $\sigma_{1}=\sigma_{2}$. Thus, the Equation (Equation 21) is valid if and only if
$$\begin{array}{c} {\left|x\right|}^{\delta_{1}}-{\left|x\right|}^{\delta_{2}}=0, \end{array}$$
which only happens when $\delta_{1}=\delta_{2}$, for any x∈R.

The ML estimates $\hat{\mu },\hat{\sigma },\hat{\delta }$ are the solution of the system of likelihood equations
(22)$$\begin{array}{ccc} \frac{\partial l(\Theta ;x)}{\partial \mu } & = & -\frac{n}{\hat{\sigma }}+\frac{1}{\hat{\sigma }}\sum_{i=1}^{n}e^{-\frac{x_{i}{\left|x_{i}\right|}^{\hat{\delta }}+\hat{\mu }}{\hat{\sigma }}}=0; \end{array}$$
(23)$$\begin{array}{c} \frac{\partial l(\Theta ;x)}{\partial \sigma }=-\frac{n}{\hat{\sigma }}-\sum_{i=1}^{n}{\hat{\sigma }}^{-2}(x_{i}|x_{i}{|}^{\hat{\delta }}+\hat{\mu })-\sum_{i=1}^{n}{\hat{\sigma }}^{-2}(x_{i}|x_{i}{|}^{\hat{\delta }}+\hat{\mu })e^{-\left(x_{i}{\left|x_{i}\right|}^{\hat{\delta }}+\hat{\mu }\right){\hat{\sigma }}^{-1}}=0; \end{array}$$
(24)$$\begin{array}{cc} \frac{\partial l(\Theta ;x)}{\partial \delta } & =\frac{n}{\hat{\delta }+1}+\sum_{i=1}^{n}\ln\left|x_{i}\right|-\frac{1}{\hat{\sigma }}\sum_{i=1}^{n}x_{i}{\left|x_{i}\right|}^{\hat{\delta }}\ln\frac{\sum_{k=1}^{n}x_{k}{\left|x_{k}\right|}^{\hat{\delta }}}{\hat{\sigma }}\\ & +e^{\frac{\hat{\mu }}{\hat{\sigma }}}\sum_{i=1}^{n}e^{-\frac{x_{i}{\left|x_{i}\right|}^{\hat{\delta }}}{\hat{\sigma }}}\left(\frac{\sum_{j=1}^{n}x_{j}{\left|x_{j}\right|}^{\hat{\delta }}}{\hat{\sigma }}\right)\ln\frac{\sum_{k=1}^{n}x_{k}{\left|x_{k}\right|}^{\hat{\delta }}}{\hat{\sigma }}=0. \end{array}$$

After algebraic manipulations, we get the unique closed-form solution for estimating μ,
(25)$$\begin{array}{c} \hat{\mu }=\hat{\sigma }\ln\frac{\sum_{i=1}^{n}e^{-\frac{x_{i}{\left|x_{i}\right|}^{\hat{\delta }}}{\hat{\sigma }}}}{n}. \end{array}$$
However, the estimates $\hat{\sigma }$ and $\hat{\delta }$ must be obtained numerically.

### Numerical Performance of ML Estimates

Now, we perform a Monte Carlo study to assess the performance of maximum likelihood estimators $\hat{\mu }$, $\hat{\sigma }$ and $\hat{\delta }$ in terms of their means, mean squared errors (MSE), biases, and standard errors (SE). We defined a set of 9 parameter vectors, $\theta_{1},\ldots ,\theta_{9}$, with μ∈{−1,0,1}, δ∈{0,2,4}, and σ=1, for three sampling scenarios: n=50,100, and 1000. For each of the 27 combinations between parameters and sample scenarios, we took 100 Monte Carlo replications using the software R (version 3.4.1) to get the empirical sampling distributions of $\hat{\mu }$, $\hat{\sigma }$, and $\hat{\delta }$. We generate the Monte Carlo variates of $X∼F_{GB}\left(\cdot ;\mu ,\sigma ,\delta \right)$ through the inverse transform method with the quantile Function (Equation 17).

Table 1, Table 2 and Table 3 depict the empirical expected values, bias, MSE, and SE of the ML estimators of the IBG model. Figure 4, Figure 5 and Figure 6 illustrate the empirical behavior of the MSE vs *n*. Overall, the MSE decreases as the sample size increases, confirming the optimal properties of ML estimators from the statistical inference theory. In this study, we did not face numerical problems in estimating these parameters.

## 4. Application

We use two financial data sets taken from https://finance.yahoo.com (accessed on 1 December 2021) to illustrate the applicability of the invertible bimodal Gumbel model. The first is the daily stock prices of Petrobras (PETR4), quoted in US dollars, from 1 March 2000 to 10 January 2021, totaling 5465 observations. The other is the daily exchange rate of the Brazilian real against the US dollar (USD/BRL) from 12 January 2003 to 15 October 2021, totaling 4223 data points. We aim to get the value-at-risk (VaR) of these data, a common measure of financial risk. It denotes the maximum loss incurred on a portfolio over a specific time horizon with a given confidence level 1−α [22]. It is expressed in probabilistic terms as
(26)$$\begin{array}{c} {\mathrm{VaR}}_{\alpha }\left(X_{t}\right)=\inf\left\{x\in R:F\left(x\right)\geq \alpha \right\}, \end{array}$$
where F(x) is the cumulative distribution function (CDF) of a real random variable $X_{t}$ observed at time t∈{0,1,2,…}, and 0<α<1 is a small prespecified probability. Particularly, in our study, the time horizon comprises the totality of data in each discrete time series. Moreover, as $X_{t}$ is an absolutely continuous random variable with an invertible CDF, we can write
$$\begin{array}{ccc} x_{\alpha } & = & {\mathrm{VaR}}_{\alpha }\left(X_{t}\right)=F^{-1}\left(1-\alpha \right), \end{array}$$
where $F^{-1}$ denotes the inverse function of *F*, and $x_{\alpha }$ is the *α*—quantile of $X_{t}$. As usual, $X_{t}$ means the log return of prices, that is,
(27)$$\begin{array}{c} X_{t}=\ln P_{t}-\ln P_{t-1}, \end{array}$$
where $P_{t}$ is a price at time *t*.

Table 4 summarizes descriptive statistics for PETR4 and USD/BRL returns. The return averages are close to zero, and the proximity between the absolute values of the first and third quartiles indicates the possible symmetry of the data, except for the possible extreme values suggested by the maximum (PETR4) and minimum (USD/BRL) statistics. Indeed, Figure 7 and Figure 8 depict extreme returns, some of them due to the COVID-19 event. In the critical period of the pandemic, Petrobras shares plummeted 57% due to low demand for petroleum products (Figure 7). As for the exchange rate USD/BRL, the effect was the opposite: the American dollar became more expensive than the Brazilian real because of various political and economic reasons. Furthermore, in these data sets, we observe extreme positive (PETR4) and negative (USD/BRL) values that stand out significantly from the rest of the observations.

Now, we perform the block maxima and minima method to extract extreme values from our data. Let $X_{1},\cdots X_{n}$ be a random sample of log returns following $X∼F_{IBG}$. Based on its realized values ${\left\{x_{t}\right\}}_{t=1}^{n}$, we organize it into *T* non-overlapping sub-samples of length *N*, where *T* means the integer part of n/N, resulting in *T* data blocks of size *N*. We choose *N* to cover natural periods (e.g., a week or month) so that the new sub-sample is IID. Now, we take the maximum and the minimum over each *N*-history. We define the *j*th sub-sample of maximum and minimum as
(28)$$\begin{array}{c} M_{j}=\max\left\{x_{(j-1)N+1},\cdots ,x_{jN}\right\}, \end{array}$$
and
(29)$$\begin{array}{c} m_{j}=\min\left\{x_{(j-1)N+1},\cdots ,x_{jN}\right\}, \end{array}$$
for j=1,…,T. This results in a new sample of size 2T, consisting of maxima and minima,
(30)$$\begin{array}{c} {\left\{Y_{t}\right\}}_{t=1}^{2T}={\left\{m_{t},M_{t}\right\}}_{t=1}^{T}. \end{array}$$

For our case, N=15 is a block length providing IID sub-samples based on the Ljung–Box test for serial independence with a significance level of 5%. The left panels of Figure 9 and Figure 10 depict the series of extreme PETR4 and USD/BRL returns extracted from the blocks, while the left ones show the bimodal form of their distributions.

Thus, we fit these empirical distributions of extreme returns using our suggested invertible bimodal Gumbel model, $F_{IBG}\left(x;\theta \right)$, with parameters μ, σ, and δ. Table 5 shows their maximum likelihood estimates. Figure 11 depicts the fitted model against the corresponding distribution of the extracted extreme returns, indicating that the IBG is suitable for simultaneously describing minima and maxima extreme returns.

Finally, Table 6 presents the estimated VaR for α=10%, 5%, and 1%. As we are dealing with the logarithmic returns, to make these VaR values more understandable, we may consider that the maximum return of the stock is expVaR−1. Thus, for example, over a 15-day period, we do not expect a return greater than 7.4% for PETR4 and 2.6% for USD/BRL with a confidence of 90%.

## 5. Conclusions

This paper introduced and examined the IBG distribution as an extension of the classical Gumbel distribution. We addressed the limitations of the unimodal Gumbel by proposing a model capable of simultaneously representing both maximum and minimum extremes, enhancing its applicability and versatility. The mathematical formulations accompanied by illustrative figures elucidate the characteristics and behavior of the proposed distribution, emphasizing its advantages in terms of computational efficiency and flexibility. We presented its distributional properties, including mode, moment-generating functions, and entropy. In our illustration, we performed the maximum and minimum blocks technique to obtain serial independent data for the VaR estimation through the ML method, offering a novel perspective on modeling extremes through the lens of the invertible bimodal Gumbel distribution.

## Figures and Tables

**Figure 1 entropy-25-01598-f001:**
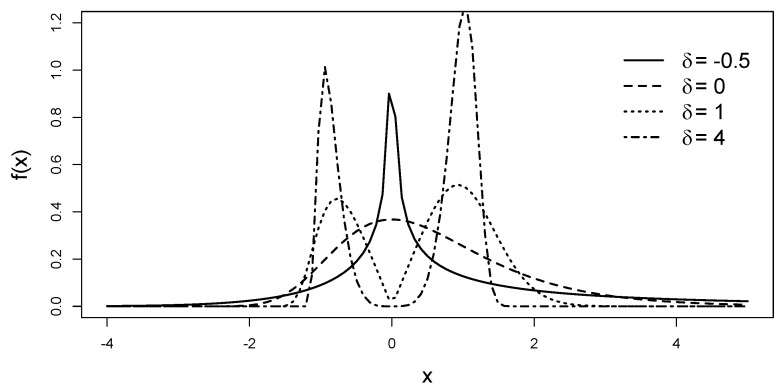
Density $f_{\mathrm{IBG}}\left(x;0,1,\delta \right)$, with δ ranging from −0.5 to 4. Bimodal distributions appear when δ>0, and we have the unimodal Gumbel if δ=0.

**Figure 2 entropy-25-01598-f002:**
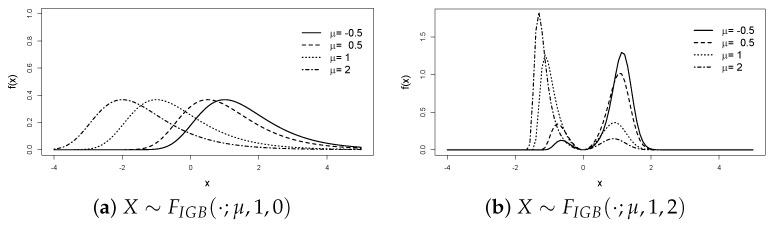
Density $f_{\mathrm{IBG}}\left(\cdot ;\mu ,1,\delta \right)$, with μ ranging from − 0.5 to 2, δ=0 or 1, and μ acting as a location (**a**) or shape parameter (**b**).

**Figure 3 entropy-25-01598-f003:**
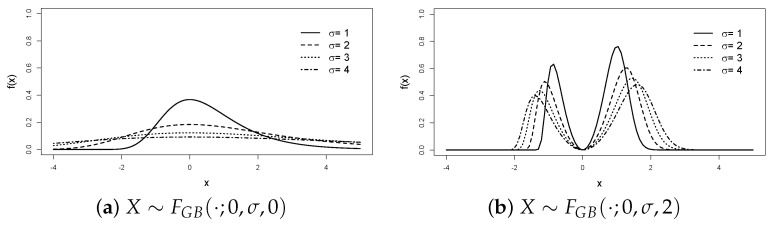
Density $f_{\mathrm{IBG}}\left(\cdot ;0,\sigma ,\delta \right)$, with σ ranging from 1 to 4 and δ=0 (unimodal, (**a**)) or 2 (bimodal, (**b**)). In both cases, σ represents the scale parameter.

**Figure 4 entropy-25-01598-f004:**
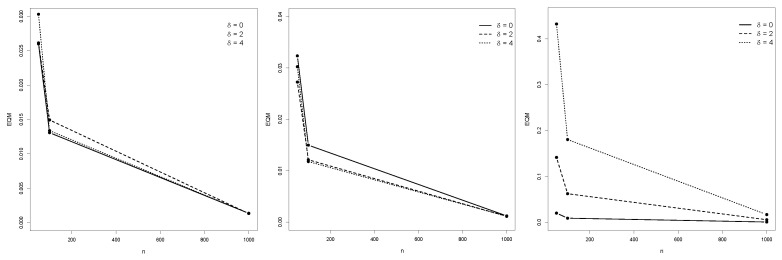
MSE from Monte Carlo replications of samples with *n* ranging from 50 to 1000, for ${\hat{\theta }}_{1}$, ${\hat{\theta }}_{2}$, and ${\hat{\theta }}_{3}$.

**Figure 5 entropy-25-01598-f005:**
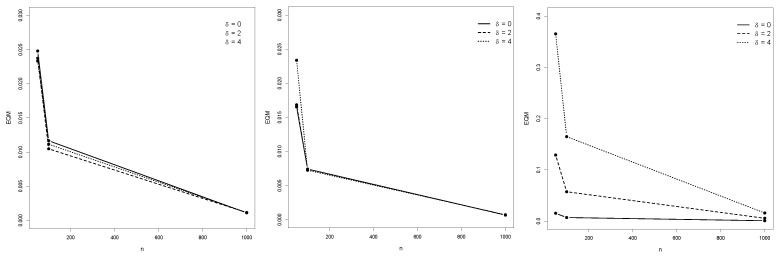
MSE behavior of ${\hat{\theta }}_{4},\;{\hat{\theta }}_{5}$ and ${\hat{\theta }}_{6}$. MSE from Monte Carlo replications of samples with *n* ranging from 50 to 1000, for ${\hat{\theta }}_{4}$, ${\hat{\theta }}_{5}$, and ${\hat{\theta }}_{6}$.

**Figure 6 entropy-25-01598-f006:**
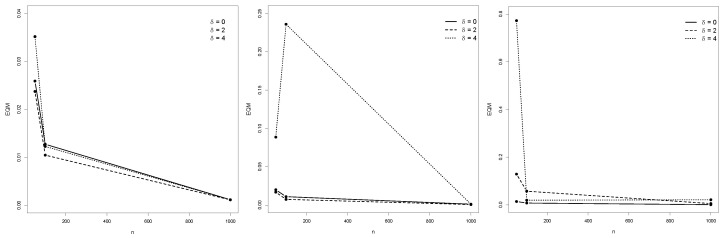
MSE from Monte Carlo replications of samples with *n* ranging from 50 to 1000, for ${\hat{\theta }}_{7}$, ${\hat{\theta }}_{8}$, and ${\hat{\theta }}_{9}$.

**Figure 7 entropy-25-01598-f007:**
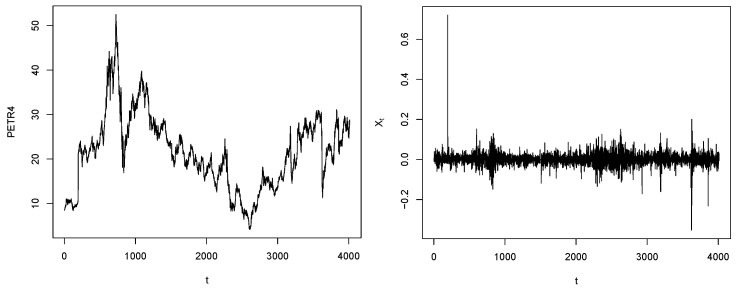
PETR4 Prices (**left**) and PETR4 log returns (**right**).

**Figure 8 entropy-25-01598-f008:**
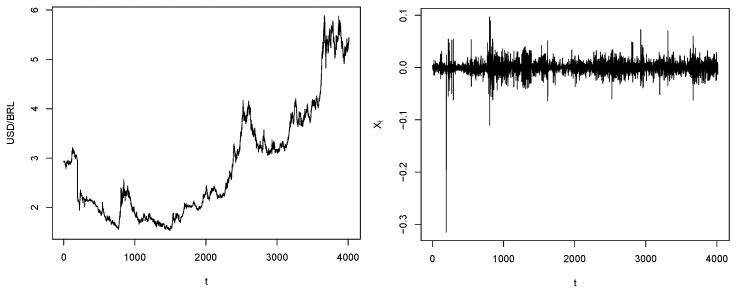
USD/BRL Prices (**left**) and USD/BRL log returns (**right**).

**Figure 9 entropy-25-01598-f009:**
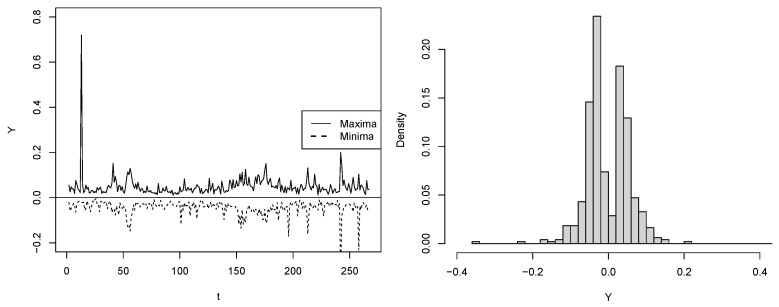
PETR4 sub-sample ${\left\{Y_{t}\right\}}_{t=1}^{70}$: Extremes obtained from the blocks (**left**) and histogram (**right**).

**Figure 10 entropy-25-01598-f010:**
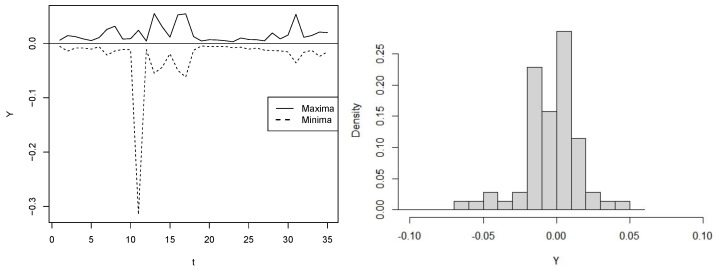
USD/BRL sub-sample ${\left\{Y_{t}\right\}}_{t=1}^{70}$: Extremes obtained from the blocks (**left**) and histogram (**right**).

**Figure 11 entropy-25-01598-f011:**
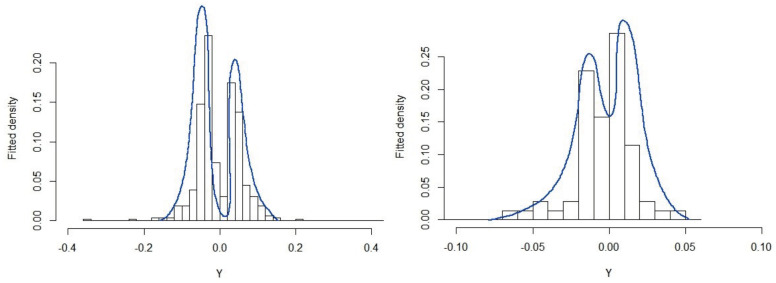
Histogram *versus* fitted distribution: PETR4 (**left**) and USD/BRL (**right**).

**Table 1 entropy-25-01598-t001:** Means, biases, mean squared errors (MSE), and standard errors (SE) of the estimated parameters from 100 Monte Carlo replications of samples with n=50.

	θ	True	Mean	Bias	MSE	SE
	μ	−1	−1.00952	−0.00952	0.02163	0.1475
$\theta_{1}$	σ	1	0.97492	−0.02507	0.02765	0.1653
	δ	0	0.01114	0.01114	0.01554	0.1248
	μ	−1	−1.03142	−0.03142	0.03419	0.1831
$\theta_{2}$	σ	1	1.02366	0.02366	0.02592	0.1600
	δ	2	2.05863	0.05863	0.12062	0.3440
	μ	−1	−0.99788	0.00211	0.02748	0.1666
$\theta_{3}$	σ	1	0.96513	−0.03486	0.02763	0.1634
	δ	4	3.91971	−0.08028	0.34339	0.5834
	μ	0	0.00145	0.00145	0.02113	0.1461
$\theta_{4}$	σ	1	0.99965	−0.00034	0.01646	0.1289
	δ	0	0.03047	0.03047	0.01316	0.1111
	μ	0	0.00339	0.00339	0.02363	0.1544
$\theta_{5}$	σ	1	1.00572	0.00572	0.01401	0.1188
	δ	2	2.14749	0.14749	0.14933	0.3589
	μ	0	−0.02841	−0.02841	0.02129	0.1438
$\theta_{6}$	σ	1	0.97091	−0.02908	0.01374	0.1141
	δ	4	4.00093	0.00093	0.26488	0.5172
	μ	1	0.95417	−0.04582	0.02859	0.1636
$\theta_{7}$	σ	1	0.98530	−0.01469	0.01918	0.1384
	δ	0	0.00560	0.00560	0.01242	0.1118
	μ	1	0.96207	−0.03792	0.02218	0.1447
$\theta_{8}$	σ	1	1.04081	0.04081	0.02338	0.1481
	δ	2	2.10659	0.10659	0.17084	0.4013
	μ	1	0.93990	−0.06009	0.05954	0.2376
$\theta_{9}$	σ	1	0.97762	−0.02237	0.12103	0.3489
	δ	4	3.96419	−0.03580	0.91265	0.9594

**Table 2 entropy-25-01598-t002:** Means, biases, mean squared errors (MSE), and standard errors (SE) of the estimated parameters from 100 Monte Carlo replications of samples with n=100.

	θ	True	Mean	Bias	MSE	SE
	μ	−1	−0.98025	0.01974	0.01280	0.1120
$\theta_{1}$	σ	1	0.98180	−0.01819	0.01146	0.1060
	δ	0	−0.00898	−0.00898	0.00780	0.0883
	μ	−1	−0.99734	0.00265	0.01532	0.1243
$\theta_{2}$	σ	1	0.99027	−0.00972	0.01188	0.1091
	δ	2	2.01504	0.01504	0.06759	0.2608
	μ	−1	−0.99937	0.00062	0.01558	0.1254
$\theta_{3}$	σ	1	0.97695	−0.02304	0.01185	0.1069
	δ	4	3.9113	−0.08860	0.13284	0.3553
$\theta_{4}$	μ	0	−0.02087	−0.02087	0.01232	0.1095
	σ	1	0.99485	−0.00514	0.00724	0.0854
	δ	0	0.02235	0.02235	0.00722	0.0824
	μ	0	−0.01413	−0.01413	0.01182	0.1083
$\theta_{5}$	σ	1	0.98941	−0.01058	0.00694	0.0830
	δ	2	2.06691	0.06691	0.07194	0.2610
	μ	0	−0.00765	−0.00765	0.01211	0.1103
$\theta_{6}$	σ	1	0.99801	−0.00198	0.00713	0.0848
	δ	4	4.05904	0.05904	0.16391	0.4025
	μ	1	0.94706	−0.05293	0.01350	0.1039
$\theta_{7}$	σ	1	0.98666	−0.01333	0.00964	0.0977
	δ	0	−0.02825	−0.0282	0.00588	0.0717
	μ	1	0.96052	−0.03947	0.014831	0.1157
$\theta_{8}$	σ	1	1.01628	0.01628	0.01126	0.1053
	δ	2	1.99650	−0.00349	0.06319	0.2526
	μ	1	0.98981	−0.01018	0.01102	0.1050
$\theta_{9}$	σ	1	0.99397	−0.00602	0.00709	0.0844
	δ	4	4.02093	0.02093	0.11929	0.3464

**Table 3 entropy-25-01598-t003:** Means, biases, mean squared errors (MSE), and standard errors (SE) of the estimated parameters from 100 Monte Carlo replications of samples with n=1000.

	θ	True	Mean	Bias	MSE	SE
	μ	−1	−0.99837	0.00162	0.00149	0.0388
$\theta_{1}$	σ	1	0.99413	−0.00586	0.00104	0.0319
	δ	0	−0.00209	−0.00209	0.00072	0.0269
	μ	−1	−0.99906	0.00093	0.00130	0.0362
$\theta_{2}$	σ	1	1.00507	0.00507	0.00088	0.0293
	δ	2	2.01457	0.01457	0.00540	0.0724
	μ	−1	−0.99417	0.00582	0.00110	0.0328
$\theta_{3}$	σ	1	0.99574	−0.00425	0.00091	0.03016
	δ	4	3.99378	−0.00621	0.01433	0.1201
	μ	0	−0.00232	−0.00232	0.00134	0.0367
$\theta_{4}$	σ	1	1.00226	0.00226	0.00069	0.02643
	δ	0	0.00303	0.00303	0.00076	0.0277
	μ	0	−0.00322	−0.00322	0.00094	0.0307
$\theta_{5}$	σ	1	0.99633	−0.00366	0.00062	0.0248
	δ	2	2.01715	0.01715	0.00587	0.0751
	μ	0	−0.00824	−0.00824	0.00114	0.0330
$\theta_{6}$	σ	1	1.00379	0.00379	0.00065	0.0254
	δ	4	4.01962	0.01962	0.01519	0.1223
	μ	1	0.99310	−0.00689	0.00136	0.0365
$\theta_{7}$	σ	1	0.99649	−0.00350	0.00107	0.0327
	δ	0	−0.00116	−0.00116	0.00080	0.0284
	μ	1	0.95857	−0.04142	0.00253	0.0286
$\theta_{8}$	σ	1	0.99516	−0.00483	0.00104	0.0320
	δ	2	1.96530	−0.03469	0.00929	0.0904
	μ	1	0.99638	−0.00361	0.00133	0.0365
$\theta_{9}$	σ	1	1.00135	0.00135	0.00108	0.0330
	δ	4	4.00819	0.00819	0.01701	0.1308

**Table 4 entropy-25-01598-t004:** Descriptive statistics.

Stock	Minimum	1st Quartile	Median	Mean	3rd Quartile	Maximum
PETR4	−0.3523667	−0.0137843	0.0000000	0.0003036	0.0138190	0.7203695
USD/BRL	−0.3148314	−0.0056746	0.0000000	0.0001515	0.0060800	0.0966945

**Table 5 entropy-25-01598-t005:** ML estimates of θ=(μ,σ,δ) and their respective standard error (SE).

Stock	$\hat{\mu }$	$\hat{\sigma }$	$\hat{\delta }$
PETR4	0.000009962	0.000089877	1.246255323
SE	0.0000071	0.0000003	0.0000083
USD/BRL	0.000016486	0.000099908	1.31295954
SE	0.0000007	0.0000001	0.0000027

**Table 6 entropy-25-01598-t006:** Estimated ${\mathrm{VaR}}_{\alpha }$ from (Equation 17).

Stock	10%	5%	1%
PETR4	0.07166153	0.07835825	0.09003984
USD/BRL	0.02594245	0.0295429	0.03612925

## Data Availability

https://finance.yahoo.com, accessed on 1 December 2021.

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
