# Peer review of "Bridging Extremes: The Invertible Bimodal Gumbel Distribution"

_entropy, 2023, doi:10.3390/e25121598_

Round 1

Reviewer 1 Report

Comments and Suggestions for Authors The authors introduce a bimodal invertible Gumbel distribution. The article is not properly organized; two applications to financial data are shown.The following points are suggestions for the authors to improve the manuscript.1) They should better explain what is meant by bimodal heavy-tailed distributions.2) The IBG distribution is slightly flexible when $x=0$; moreover it is non-differentiable at this point.3) They should discuss the existence of ML estimators.4) In the Application, they should include the standard errors of the ML estimates.

Author Response

We thank you reviewer for your careful reading and comments on our paper. We are delighted that all comments were positive, and we have prepared the revision taking into account all these comments. We do think that the revised manuscript represents a very good improvement of our work. We now answer all your comments. All new materials are in red color 

Reviewer #1: 1) They should better explain what is meant by bimodal heavy-tailed distributions.

Response: Thank you for your suggestion. We added the following in the Introduction (lines 15 - 18): “Two main features characterize it. Firstly, it exhibits bimodality, meaning it has two distinct peaks or modes, indicating the presence of two prominent regimes within the overall data set. Secondly, it has heavy tails, which means a higher likelihood of occurrences of extreme values than light tails distributions.”

Reviewer #1: 2) The IBG distribution is slightly flexible when $x=0$; moreover it is non-differentiable at this point.

Response: Yes, indeed. However, it is not a problem as a continuous distribution because this point occurs with zero probability, as with, for example, the Laplace distribution. Thus, this fact does not affect our results.

Reviewer #1: 3) They should discuss the existence of ML estimators.

Response: Thank you for this. We added the following just after equation (20) (lines 133 - 144): “This log-likelihood function is well-defined across the entire parameter space and is continuous and differentiable for the vector parameters. Additionally, the family of distributions FIBG is identifiable, meaning different parameters should lead to distinct probability distributions, ensuring a unique maximum for the likelihood function.”

We also discuss that FIBG is identifiable at lines 127-144

Reviewer #1: 4) In the Application, they should include the standard errors of the ML estimates.

Response: We agree. Now, Table 5 shows the standard errors.

Reviewer 2 Report

Comments and Suggestions for Authors

In the article, the authors propose applying the invertible bimodal Gumbel (IBG) distribution, which allows simultaneous modeling of maximum and minimum extremes, to data from financial markets. The article is quite well done from a methodological and statistical point of view, but it needs several corrections and additions, which I have listed below.

1. The article has a weak theoretical foundation in terms of complexity science. The distribution studied in the paper is simply one exemplification of the catastrophe theory developed by René Thom, which includes probability distributions of this type. Examples include such elementary catastrophes as cusp and butterfly, which generate distributions very similar to those in Figures 1-3 and 9-11. The theoretical aspects of the work should be strengthened by referring to this concept. This is a certain theoretical minimum that should be included in the paper. Of course, there are other strands of theoretical research referring to the distribution proposed by the authors.

2. The authors used the IBG distribution to estimate a well-known measure of financial risk - value at risk (VaR). However, the results obtained are suspended in a vacuum, as they cannot be related to VaR estimations made using other statistical methods. This definitely lowers the practical dimension of the research and means that the goals of the work have not been fully achieved. In other words, the usefulness of the IBG distribution for improving risk management strategies has not been proven. This contradicts the claims made in the abstract.

3. In Figures 9-11, the coordinate axes are not signed.

Author Response

We thank you reviewer for your careful reading and comments on our paper. We are delighted that all comments were positive, and we have prepared the revision taking into account all these comments. We do think that the revised manuscript represents a very good improvement of our work. We now answer all your comments. All new materials are in red.

Reviewer #2: 1) The article has a weak theoretical foundation in terms of complexity science. The distribution studied in the paper is simply one exemplification of the catastrophe theory developed by René Thom, which includes probability distributions of this type. Examples include such elementary catastrophes as cusp and butterfly, which generate distributions very similar to those in Figures 1-3 and 9-11. The theoretical aspects of the work should be strengthened by referring to this concept. This is a certain theoretical minimum that should be included in the paper. Of course, there are other strands of theoretical research referring to the distribution proposed by the authors..

Response: That is a good point. We added the following in the Introduction (lines 19 - 32): “Bimodal heavy-tailed distributions are related to René Thom's catastrophe theory, focusing on systems characterized by sudden, dramatic changes and a propensity for extreme events [e.g. 3, 7]. Catastrophe theory deals with situations where small parameter changes can lead to abrupt shifts in the system's state. This concept aligns with bimodal distributions, where a system may switch between two states or regimes. The heavy-tailed aspect of these distributions reflects the likelihood of rare, extreme events, mirroring the focus of catastrophe theory on significant, sudden changes. Both concepts encapsulate unpredictability and uncertainty inherent in the systems they describe. Catastrophe theory provides a mathematical framework for understanding these dynamics, which can manifest statistically as bimodal, heavy-tailed distributions. This connection is especially relevant in economics and finance, where catastrophic shifts and heavy-tailed distributions are frequently observed. Essentially, the interplay between these concepts helps understand and model systems where small inputs or changes can lead to significant, unpredictable, and often extreme outputs or shifts.”

Reviewer #2: 2) The authors used the IBG distribution to estimate a well-known measure of financial risk - value at risk (VaR). However, the results obtained are suspended in a vacuum, as they cannot be related to VaR estimations made using other statistical methods. This definitely lowers the practical dimension of the research and means that the goals of the work have not been fully achieved. In other words, the usefulness of the IBG distribution for improving risk management strategies has not been proven. This contradicts the claims made in the abstract.

Response: We agree. We soften this point and now, the last sentence in the abstract reads as follows: “We illustrate using financial data to estimate Value at Risk (VaR) from our suggested model, considering maximum and minimum blocks simultaneously.”

Reviewer #2: 3) In Figures 9-11, the coordinate axes are not signed.

Response: We agree. We corrected these figures.